# Recent Advances in Nanomaterials of Group XIV Elements of Periodic Table in Breast Cancer Treatment

**DOI:** 10.3390/pharmaceutics14122640

**Published:** 2022-11-29

**Authors:** Azadeh Hekmat, Luciano Saso, Viney Lather, Deepti Pandita, Irena Kostova, Ali Akbar Saboury

**Affiliations:** 1Department of Biology, Science and Research Branch, Islamic Azad University, Tehran 1477893855, Iran; 2Department of Physiology and Pharmacology “Vittorio Erspamer”, Sapienza University, I-00185 Rome, Italy; 3Amity Institute of Pharmacy, Amity University, Noida 201313, Uttar Pradesh, India; 4Department of Pharmaceutics, Delhi Institute of Pharmaceutical Sciences & Research (DIPSAR) Delhi Pharmaceutical Sciences and Research University, Pushp Vihar, Government of NCT of Delhi, New Delhi 110017, India; 5Centre for Advanced Formulation and Technology (CAFT), Delhi Pharmaceutical Sciences and Research University, Pushp Vihar, Government of NCT of Delhi, New Delhi 110017, India; 6Department of Chemistry, Faculty of Pharmacy, Medical University, 2 Dunav St., 1000 Sofia, Bulgaria; 7Institute of Biochemistry and Biophysics, University of Tehran, Tehran 1417466191, Iran

**Keywords:** group XIV, breast cancer, nanopharmaceuticals, graphene nanomaterials, diamond nanomaterials, lead nanomaterials

## Abstract

Breast cancer is one of the most common malignancies and a leading cause of cancer-related mortality among women worldwide. The elements of group XIV in the periodic table exhibit a wide range of chemical manners. Recently, there have been remarkable developments in the field of nanobiomedical research, especially in the application of engineered nanomaterials in biomedical applications. In this review, we concentrate on the recent investigations on the antiproliferative effects of nanomaterials of the elements of group XIV in the periodic table on breast cancer cells. In this review, the data available on nanomaterials of group XIV for breast cancer treatment has been documented, providing a useful insight into tumor biology and nano–bio interactions to develop more effective nanotherapeutics for cancer patients.

## 1. Introduction

Cancer is a prominent cause of death worldwide. In cancer, cells with uncontrolled proliferation spread from an original focal point to other parts of the body and cause death. The four most widespread forms of cancer are lung, colorectal, prostate, and breast cancer. Breast cancer is a type of malignancy that originates in the breast cells and is the second leading cause of death in women [1,2]. According to the American Cancer Society’s estimates for 2022, nearly 287,850 new cases of invasive breast cancer will be identified in women in the United States and, unfortunately, nearly 43,250 women will die from this type of cancer. Therefore, it is crucial to ensure earlier detection and treatment of breast cancer to diminish disease spread and mortalities [1].

In clinical practice, several kinds of targeted nanomaterials such as Doxil^®^, Myocet^®^, and Abraxane^®^ are accessible for breast cancer therapy [3]. Recently, there have been remarkable developments in the field of nanopharmacological research. The size and surface charge of nanomaterials, the encapsulation capacity of the nanomaterials, long half-life in circulation, high drug-loading efficiency, minimum systemic toxicity, high adhesion to the tumor environment, selective localization, and improved internalization into the tumor cells are the crucial factors for utilizing nanomaterials in cancer treatment [4]. Presently, there are numerous FDA (Food and Drug Administration)-approved nanoparticles (NPs) for clinical use [5].

One category of nanomaterials used in cancer treatment belongs to the elements of group XIV (G14) in the periodic table. All the elements in the G14 have four electrons in their outer shell. The elements of G14 demonstrate a broad variety of physicochemical and pharmacological properties. This group consists of carbon (C), silicon (Si), germanium (Ge), tin (Sn), lead (Pb), and flerovium (Fl). Numerous experiments have analyzed the effects of these types of nanomaterials, concluding that the application of these nanomaterials in cancer drug design can induce a balance between enhancing efficacy and decreasing the toxicity of drugs. With the rapid improvement of nanotechnology, this paper will review and concentrate on the recent (<5 years) investigations about the application of nanomaterial derived from G14 (Figure 1) in breast cancer treatment, with a focus on their advantages and limitations during use.

## 2. Breast Cancer Subtypes

Breast cancer is a heterogeneous disease with various subtypes and diverse biological characteristics that lead to different responses to clinical treatments. More than 75% of breast cancers are positive for estrogen receptors (ER) and/or progesterone receptors (PR). The ER-positive tumors express genes that encode typical proteins of luminal epithelial cells; therefore, they are named the luminal group. According to the combined data set, two main luminal-like subclasses were identified: luminal A and luminal B [6]. The other subtypes of breast cancer are HER2-positive, claudin-low, and basal-like (triple-negative) subtypes.

Luminal A tumors are marked by a high level of ER (ER-positive) and/or PR (PR-positive), a low level of Ki67 (proliferating cell nuclear antigen) index, as well as negative human epidermal growth factor receptor-2 (HER2) expression. They are also characterized by the expression of luminal epithelial cytokeratins (CK) 18 and 8 and other luminal-associated markers. The luminal A subtype represents about 50–60% of all breast cancers and commonly has a low degree of nuclear pleomorphism, low mitotic activity, and low histological grade. Of the five major subtypes, luminal A tumors possess a good prognosis (chance of survival) and relatively low recurrence rates [6].

Luminal B tumors tend to be ER-positive; however, they can be HER2-negative or HER2-positive. Luminal B tumors are marked by a high level of growth receptor signaling genes. Compared to the luminal A subtype, this subtype has a higher histological grade, more aggressive phenotype, worse prognosis, higher level of proliferation-related gene expression, higher recurrence rate, and lower survival rates after relapse. The luminal B subtype represents about 15–20% of all breast cancers [6,7].

The HER2-enriched subtype is characterized by a high level of HER2 and other genes associated with the HER2 pathway and/or HER2 amplicon expression. HER2 is a one of a family of four membrane tyrosine kinases. The HER2 receptor (a proto-oncogene mapped in chromosome 17q21) is encoded by the HER2 gene. Approximately half of the HER2-enriched breast cancers are positive for ER, although they generally express a lower level of ER. HER2-enriched tumors represent about 15–20% of all breast cancers. This subtype has about 40% p53 mutations, high histological and nuclear grade, and a high degree of proliferative activity. In the absence of treatment, HER2-enriched tumors have a poor prognosis [6,7,8].

Basal-like subtype tumors express high levels of basal myoepithelial markers but do not express HER2, PR, and ER; consequently, they are referred to as triple-negative [9]. Basal-like tumors are marked by aggressive clinical behavior, a high rate of metastasis to the lung and brain, high proliferative indices and mitosis, the presence of central necrotic or fibrotic zones, and poor tubule formation. Tumors belonging to this subtype overexpress epidermal growth factor receptor (EGFR), alpha-beta crystallin, caveolins 1 and 2, fascin, and P-cadherin. Patients with basal-like tumors have poor clinical outcomes. It is crucial to explain that there is an 80% overlap between the intrinsic basal-like and triple-negative subtypes. The basal-like subtype is defined through gene expression microarray analysis; nevertheless, the term triple-negative belongs to the immunohistochemical classification of breast cancers lacking HER2, PR, and ER protein expression [6,7].

Claudin-low tumors are described by a low level of expression of genes involved in cell–cell adhesions and tight junctions (including E cadherin, occluding, and claudins 3, 4, and 7), a high level of epithelial to mesenchymal transition gene expression, and zero level of HER2, PR, and ER expression. it has been confirmed that patients with claudin-low tumors have poor clinical outcomes [6,7].

According to current studies, each subtype has a different treatment response. Since ER is a therapeutic target, the luminal A and luminal B subtypes can be treated with hormone therapy. The HER2-enriched tumors are potential candidates for trastuzumab therapy. However, basal-like tumors are not easy to treat and regularly have a poor prognosis [7].

## 3. Breast Cancer Chemotherapy

Depending on the cancer subtype, there are various accessible treatment strategies: (1) surgery, (2) hormone therapy, (3) radiotherapy, (4) antibody-based therapy, and (5) chemotherapy [8]. In chemotherapy, malignancy is cured by chemical drugs, which have the capability to block cell proliferation and induce cell apoptosis. All breast carcinomas do not require the application of chemotherapy; however, it is essential in certain circumstances, such as before (neoadjuvant chemotherapy) or after surgery (adjuvant chemotherapy), or in advanced tumors where chemotherapy could be a key cancer treatment strategy. The application of cytotoxic chemotherapy in both advanced and early stage breast cancer has made meaningful progress in the last 20 years, with numerous landmark experiments discovering certain survival benefits for newer therapies. Chemotherapeutic drugs such as 5-fluorouracil (5-FU), paclitaxel (PTX), mitoxantrone (MTX), cisplatin, and doxorubicin (DOX) have been verified to inhibit growth rates of breast cancer cells and to limit their metabolic functions. However, the undesirable complications related to these drugs, due to their cytotoxicity toward healthy/normal cells, have reduced the application of chemotherapy in cancer therapy. The distinctive properties of nanomaterials promise to defeat some treatment limitations in conventional breast cancer therapy. As mentioned earlier, there are various nanomedicines approved for the treatment of breast cancer including Abraxane^®^, Myocet^®^, Genexol^®^, PICN^®^, and Nanoxel^®^; however, there are still limitations that should be overcome. Non-specific interaction with the biological systems, immunological interactions, biocompatibility reduction, and tumor biological particularities are the major limitations of nanodrug efficacy [10,11]. In this sense, the development of newer nanomaterial-based anticancer agents could overcome the handicaps of chemotherapeutic treatments, diminishing their toxicity and enhancing their efficiency.

## 4. Application of Breast Cancer Cell Lines for Nanopharmaceutical Studies

The term nanopharmaceuticals, a crucial part of nanomedicine, includes the application of nanomaterials as therapeutic agents as well as the discovery and delivery of drugs utilizing nanobiotechnology [12]. The development of innovative nanodrugs is an extremely expensive, complex, and time-consuming procedure. However, there are several selection strategies to examine the effectiveness of test compounds. Generally, human cancer cell lines are employed in nanopharmaceutical research for anticancer drug screening, as they characterize the origin of cancer. One of the main advantages of utilizing cultured cell lines in cancer research is that they offer a fairly homogeneous cell population that is able to self-replicate in a standard cell culture medium. They are easy to handle and can be replaced from frozen stocks if lost as a result of contamination. Moreover, they may also help in determining the cellular toxicity of the drugs [13]. In Table 1, a list of breast cancer cell lines that are mainly employed in breast cancer research is shown. Therefore, in the following section, the effects of nanomaterials of the elements of the G14 on the growth of different breast cancer cell lines will be discussed.

## 5. Carbon-Based Nanomaterials

Carbon is one of the most widespread elements in the ecosystem and carbon materials are well known for being more biologically and environmentally friendly than inorganic materials. Carbon atoms can undergo sp, sp^2^, and sp^3^ hybridizations with a narrow bandgap between their 2s and 2p electronic shells. Graphite with sp^2^ hybridization and diamond with sp^3^ hybridization are the two commonly known allotropic forms of carbon. Carbon nanomaterials have laid down their historical footprints in the scientific investigation field with the first studies on fullerenes and related compounds throughout the mid-1980s [14,15]. Currently, carbon nanomaterials including graphene, carbon nanotube (CNTs), crystalline nanodiamonds (NDs), and carbon quantum dots (CQDs) are extensively utilized in nanomedicine and nanobiotechnology due to their valuable properties. Graphite is a naturally occurring material that has been utilized in biosystems’ daily lives for hundreds of years without considerable toxicity concerns. The CNTs, graphene, NDs, and CQDs are the most considerable allotropic modifications of the nanocarbon. Each zero-dimensional (0-D) nanodiamond, one-dimensional (1-D) nanotube, and two-dimensional (2-D) graphene nanosheet can act as a prototype for the nanocomposites (Figure 2) [14].

### 5.1. Graphene-Family Nanomaterials

Graphene, one of the crystalline forms of carbon, is a monolayer of carbon atoms tightly packed into a 2D honeycomb lattice. Each carbon atom of graphene has one out-of-plane *π* bond and three *σ* bonds that can interact with neighboring atoms [16,17]. As a result of its electronic distribution and atomic structure, graphene has exciting physicochemical properties including a large surface area (about 2600 m^2^∙g^−1^), distinctive optical behaviors, and excellent chemical stability. Through physicochemical variations, graphene sheets can be converted into graphene-related materials with distinctive properties; for example, graphene oxide (GO), reduced graphene oxide (rGO), and single- and multi-layered graphene [16,17,18,19,20]. GO is usually made through exfoliation and oxidation of graphite-bearing oxygen functional groups—for example, epoxy (-O), hydroxyl (-OH), or carboxyl (-OOH)—on their edges and basal planes with the modified Hummers’ method. Microwave, thermal, chemical, microbial/bacterial, photothermal, or photochemical treatments can be applied to GO to reduce the oxygen content and initiate the production of rGO [21]. Based on the storage conditions and the preparation procedures, the surface density of rGO or GO could be altered. Due to their excellent surface functional ability, high surface area, aqueous processability, amphiphilicity, fluorescence quenching, and Raman scattering (SERS) property capacity, GO and rGO are considered promising biomaterials [22,23].

Furthermore, it has been shown that graphene could be utilized as an anticancer agent. Ribeiro et al. examined the anticancer activity of nano GO and GO modified with Poly(amidoamine) (PAMAM) and DAB-AM-16 dendrimers on MCF-7 cells. They observed that higher concentrations of nano GO induce cell death; however, lower concentrations of nano GO did not exhibit toxicity in cells. They noted that GO and modified GO particles have a great affinity with the cellular membrane [24]. Most recent cytotoxicity experiments imply that surface functionalization of GO and rGO could enhance their stability in physiological buffers as well as improve their biocompatibility. Thus, rGO- and GO-based nanocomposites could be promising nanomaterials in cancer therapy. Kodous et al. synthesized a nanocomposite consisting of rGO incorporated with Cu (rGO-Cu) and then tested the anticancer activity of the rGO–Cu nanocomposite on the MCF-7 cell line. They proved that the great inhibitory and antimetastatic effects of the rGO–Cu nanocomposite on the cells were associated with the downregulation of cathepsin D (encoded lysosomal protease protein), matrix metallopeptidase 9 (MMP-9), and B-cell lymphoma 2 (Bcl-2) gene expression and upregulation of P53 gene expression [25]. In another study, Smina et al. evaluated the anticancer activity of biogenically synthesized rGO on the MCF-7 cell line and observed the significant cytotoxic activity of their synthesized nano-rGO against breast cancer cells [26].

Currently, various rGO- and GO-based nanocomposites with antiproliferative effects have been proposed for breast cancer therapy. For instance, the anticancer effects of rGO–Fe_3_O_4_ nanocomposite on MCF-7 cells [27], rGO–Au (gold) nanocomposite on MCF-7, MDA-MB-453, and Hs 578Bst cells [28], GO–folic acid–polyethylene glycol (PEG) nanocomposite on MDAMB-231 cells [29], and GO–8-hydroxyquinoline nanocomposite on MCF-7 cells were approved [30]. Furthermore, the non-toxicity of GO- and rGO-based nanocomposites has been proven on normal cells. For example, Hekmat et al. proved that Ag nanowire–rGO composites at lower concentrations were non-toxic to human endometrial stem cells [31]. Athinarayanan et al. found the same results when they examined the effects of Cu_2_O–GO on human mesenchymal stem cell growth. They did not find any cellular damage, nuclear condensation, and DNA fragmentation in cells treated with Cu_2_O–GO [32]. Thus, rGO- and GO-based nanocomposites could be a platform for breast cancer treatment.

Most recently, researchers are turning to the development of rGO- and GO-based nanocomposites for carrying large amounts of chemotherapeutic drugs to cancer cells [33]. Quagliarini et al. demonstrated higher anticancer efficacy of DOX-loaded GO on MDA-MB 231 and MCF-7 cells. They showed that upon binding of GO to the cell plasma membrane, massive intracellular DOX can be detected [34]. Molaparast et al. also showed that biocompatible functionalized graphene nanosheets could be utilized for DOX delivery to MCF-7 cells [35]. In another study, Hatamie et al. examined the anticancer effects of curcumin-loaded rGO on MDA-MB-231 and SK-BR-3 cell lines. They observed that the curcumin–rGO sheet, at lower concentrations (<70 μg/mL), exhibited no significant toxicity, and higher concentrations of the curcumin-rGO sheet resulted in apoptosis along with a morphological transformation of the cells. Their results indicated the concentration-dependent toxicity of functionalized-rGO nanomaterials [36]. The impacts of GO–Au–5-FU nanocomposite in different molar ratios were examined on MCF-7 cells, and the results showed that the combination of 5% graphene–Au nanocomposites with 5-FU induced a higher antitumor influence and more drug stability than pure 5-FU. It seems that this nanocomposite could penetrate the cell membrane and transform liquids and then enzymes into different active metabolites [37]. Overall, it can be concluded that rGO- and GO-based nanocomposites could be promising and versatile tools for breast cancer therapy.

Among all graphene derivatives, graphene quantum dots (GQDs) are considered a newly emerging material, which represents a class of 0-D material with quasi-cyclic crystal structures. QDs could be a good source of light spanning from ultraviolet to infrared [38]. GQDs have a strong quantum property and are acquired from cutting a graphene monolayer into small pieces in dimensions of 2–20 nm. There are various reports about different applications of GQDs in medicine and pharmacy, for example in bioimaging and drug delivery, or as biosensors, and optical detectors. GQDs demonstrated good biocompatibility as well as the lowest cytotoxicity, although surface functionalization helps in their biodegradation [39,40]. The remarkable biological properties of GQDs highlight the superiority of GQDs over modified graphene or GO [41]. In the recent era, numerous structural characteristics of GQDs have been analyzed by several researchers. GQDs and graphene oxide quantum dots (GOQDs) show excellent photoluminescence properties, predominantly attributable to surface states. Accordingly, Ahirwar et al. synthesized GQDs/GOQDs (with a size range of 1.5–5.5 nm) as a photosensitizer and used 365 nm ultraviolet tube light as an irradiation source. They observed that GQDs/GOQDs were easily uptaken by cells and induced very low cytotoxicity towards MCF-7 cells. However, when 365 nm UV was performed on cells treated with GQDs/GOQDs, more than 90% of cells were killed during a short exposure of 5 min [42]. It should be noted, however, that photodynamic therapy using GQDs is very easy and effective for cancer treatment. Nevertheless, this therapy will be limited to skin-related cancers, as the depth of penetration of ultraviolet irradiation is extremely less. GQDs can also be utilized in drug delivery systems. In a recent study, GQDs and MiRGD peptides were utilized for the targeted delivery of curcumin and DOX. The in vivo study on breast cancer-bearing BALB/c mice showed that the prepared DOX-curcumin-MiRGD-GQDs peptideticles could be considered suitable multifunctional theranostic peptideticles for targeted drug delivery and tracking [43]. Liang et al. prepared pH-responsive nanoparticles loaded with GQDs and DOX (GQDs@DOX/PB). Their experiments demonstrated that GQDs@DOX/PB could promote the release of DOX in a mild acidic microenvironment [44]. Other investigations demonstrated the effective role of anticancer drugs loaded on GQD-NPs in various breast cancer models. For example, the anticancer effects of DOX-loaded zeolitic imidazolate framework GQD-NPs on 4T1 cells under near-infrared irradiation at 1.5 W/cm^2^ [45], DOX-CuS@GQD-NPs on MDA-MB-231 cells under high temperature and NIR laser irradiation at 1 W/cm^2^ [46], Herceptin-*β*-CD-GQD-HER on MCF-7 and BT-474 cells [47], Methotrexate-GQDs on MCF-7 cells [48], DOX-GQD-mesoporous silica-PEG on MDA-MB-231 [49], and DOX-*β*-cyclodextrin-SS-GQD on MDA-MB-231 and SK-BR-3 cells [50] were approved. Collectively, numerous experiments have indicated that the most invasive types of breast cancer cells can be killed efficiently with GQDs [48,49,51]. However, these experiments should be further conducted in clinical trials and animal models for realizing their potential and recognizing their safety and efficacy profile.

### 5.2. Carbon Nanotubes

CNTs were first discovered in the late 1980s. CNTs are allotropes of carbon. They are completely composed of sp^2^ carbons and can be detected as graphite sheets that have been rolled into seamless cylinders. CNTs are one of the members of the fullerene structural family categorized as multi-walled CNTs (MWCNTs) or single-walled CNTs (SWCNTs) [52,53]. Most MWCNTs are 2–100 nm in diameter, while SWCNTs are 0.4–2 nm in diameter; nevertheless, both can be millions of times longer. MWCNTs and SWCNTs naturally align themselves into cylindrical forms with diverse radiuses and “chiral” angles by means of van der Waals forces, i.e., π-stacking of sp^2^ bonds. With this property, CNTs have a unique structure and fascinating chemical and physical properties, such as high mechanical strength, excellent conductivity, and low density [52,54,55].

CNTs can be applied in numerous applications, e.g., as biodegradable polymeric scaffold materials [56] and in cancer drug delivery [57]. CNTs can act as mediators and nanocarriers for cancer treatment [58,59]. Previous analyses indicated that CNTs initiate apoptosis in cells. Few investigations have studied the cytotoxicity function of CNTs in breast cancer. Among them, some studies have found that CNTs could initiate apoptosis accompanied by inflammation [59,60]; however, other experiments indicated that CNTs could initiate apoptosis without inflammation or generate cytotoxicity through other mechanisms [61,62]. These contradictions could be attributable to the surface chemistry, purity, fabrication route, structure, concentration length, and size of CNTs. Earlier reports have indicated that the functionalization of CNTs could enhance their cytotoxicity impacts. Shaik et al. found that SWCNTs functionalized with PEG could induce a significant increase in oxidative stress and ROS formation in MDA-MB-231 cells in contrast to carboxy-functionalized SWCNTs. Furthermore, they detected that carboxy-functionalized SWCNTs were significantly safer than non-functionalized SWCNTs [63].

Recently, several experiments demonstrated that CNTs have high stability and capacity in attaching to anticancer drugs and biomolecules. It has been demonstrated that CNTs could improve considerably the effectiveness of anticancer agents with or without utilizing other treatments—for example, radiotherapy. Ünlü et al. designed a non-covalent DOX-SWCNT complex and showed that DOX-SWCNTs have higher toxic impacts on MDA-MB-231 cells in comparison to free DOX. They also indicated that long SWCNTs (length: 2.5–4 μm) induced lower cell proliferation in contrast to short SWCNTs (length: 1–1.3 μm). Their findings showed that the length of SWNTs and the exposed pH perform crucial roles in DOX loading and releasing capacity of these nanostructured materials [64]. In another study, Dan Liu et al. demonstrated that the release of DOX from a hyaluronic acid (HA)-DOX-SWNCTs complex is faster at pH 5.5 (tumor cell microenvironment) than pH 7.4 [65]. It should be noted that an HA surface-modified nanodrug delivery system can enhance the targeting of nanopreparation and relatively extend the circulation time in vivo [66]. Various other studies have demonstrated that CNTs could be considered an innovative strategy to treat breast cancer. Table 2 illustrates a brief explanation of some recent in vivo and in vitro studies that examined the therapeutic effect of CNTs complexes on breast cancer cells. According to the literature, several novel strategies for utilizing CNTs in breast cancer treatments were performed in vitro; however, some research showed that CNTs (particularly SWCNTs) at higher concentrations can lead to toxicity to organs and even cause death [59]. As an example, Alizadeh Zeinabad et al. indicated that MWCNTs, in comparison with SWCNTs, resulted in the initiation of necrotic modes of cell death. However, apoptotic modes of cell death were activated in SWCNTs-incubated cells. They proposed that the surface hydrophobicity of both membrane and CNTs is an important factor for determining absorption and interaction. In other words, interaction of MWCNTs with lower surface tension (relative to SWCNTs) and corresponding higher hydrophobicity and mitochondria membrane with higher hydrophilicity (relative to cell membrane) results in less favorable interactions and energy contribution, and corresponding agglomeration of MWCNTs (Figure 3) [53]. Hence, additional experiments must be performed to evaluate the efficacy of CNT materials in vivo. More importantly, no clinical experiments have been established based on CNTs. Thus, further investigations should evaluate the side effects of using CNTs and their complexes in breast cancer prior to using them in clinical trials.

### 5.3. Carbon Quantum Dots

Quantum dots (QDs), composed of around 100–100,000 atoms, are made from very small metal particles and in their crystal core. In ultraviolet light, these dots glow and the shade of QDs is limited by their size. [38]. Owing to their tiny size, QDs perform differently than bulk solids due to quantum-control phenomena [73]. Recently, QDs have received great attention. Some of the most interesting emerging QDs are carbon QDs (CQDs), MXenes QDs (a class of 2-D inorganic compounds that consist of carbides, nitrides, and carbonitrides), MBenes QDs (consist of 2-D transition metal borides), and metal oxide QDs (such as ZnO QDs and MgO QDs) [38]. CQDs, discovered by researchers in 2004 [74], are novel 0-D carbon-based nanomaterials known for their relatively effective fluorescence characteristics, large-scale preparation, good water solubility, small size, chemical stability, and ease of surface functionalization [74]. Various investigations have highlighted that CQDs are useful for numerous cancer treatments with or without combination with irradiation. CQDs also can effectively be utilized in drug delivery systems by combining them with antitumor drugs. For example, Li et al. designed the 5-aminolevulinic acid-CQD-glutamine-*β*-cyclodextrin system (5-ALA-CQD-Glu-*β*-CD) and loaded it with DOX. The in vitro study revealed that the produced system possessed morphological abnormalities and substantial cytotoxicity against MCF-7 cells. They also proposed that the generation of ROS upon 635 nm (for 15 min) laser irradiation improved the therapeutic effects of their designed system [75]. Samimi et al. synthesized nitrogen-doped CQD-quinic acid-gemcitabine (N-CQD-quinic acid-Gem). Then, the growth of MCF-7 cells after treatment with the synthesized system was examined. Moreover, the tissue biodistribution profiles and blood circulation of the resulting N-CQD-quinic acid-Gem were examined following intravenous administration through the tail vein of mice. The synthesized system demonstrated high tumor accumulation and great luminescent properties. They proposed that the nanosized system could penetrate the tissue system, assist drug uptake by cells, permit an effective drug delivery, and ensure action at the targeted location [76]. The cytotoxicity of DOX-transferrin-CQD on MCF-7 cells was investigated by Mahani et al. Their investigation revealed that their designed nanosystem diminished cell viability efficiently. The nanocarrier also revealed a pH-dependent DOX release behavior. The inhibitory effect of the nanocarrier could be due to its overcome of multidrug resistance of cancer cells and also targeting by conjugated transferrin on the nanosized system surface, which considerably improved the delivery of DOX into the cancer cells through the overexpressed transferrin receptors [77]. Collectively, based on previous publications, CQDs could improve drug absorption in the body and drug bioavailability, enhance drug contact criteria, and increase the medication’s impact. CQDs also could also overcome obstacles such as the solubility of the drug, lack of drug selectivity, as well as drug resistance. CQDs can interact with the cancer cells, resulting in the formation of ROS, which can kill cancer cells. However, it should be noted that traditional chemical methods make use of harsh and toxic chemical additives, which present risks to the biological environment, for the synthesis of CQDs [38,78]. Thus, various researchers tried to synthesize CQDs from biological materials. For example, Malavika et al. constructed amorphous CQDs utilizing *Aloe barbadensis miller* extract as a precursor. Their investigation showed that CQDs have apoptosis effects on MCF-7 cancer cells [79]. It has been also reported that the CQDs synthesized from ginsenoside Re [80] and walnut oil [81] were found to have anticancer properties against MCF-7 cells. Table 3 demonstrates a brief explanation of some recent studies that examined the therapeutic effect of CQDs derived from biomolecules and medicinal plants on breast cancer cells. Overall, even though CQDs have made substantial contributions to tumor therapy, numerous challenges restrict their clinical applications in humans owing to poor biocompatibility. Consequently, more biocompatible, biodegradable, and one-step synthesis methods ought to be discovered. Furthermore, a probable genotoxic effect of CQDs must be studied extensively.

### 5.4. Diamond Nanomaterials

The diamond structure is very rigid and has a tetrahedral symmetry with a strong tendency to aggregate and accumulate in aquatic environments [83]. NDs, as purely inorganic nanomaterials, are an allotrope of carbon that can be produced chemically as well as physically by high-pressure, high temperature, chemical vapor deposition, and detonation processes [84]. NDs can be categorized into three basic types: chemical vapor deposition NDs, detonation NDs, and high-pressure high-temperature NDs (HPHT) [85]. NDs are bifaceted nanomaterials possessing both diagnostic and therapeutic applications; hence, recently, they have attracted various opportunities in biomedicine and pharmacology. In a study, the anticancer property of diamond NPs (~10 nm) on MDA-MB-231cells was examined. The results showed that diamond NPs produced dose–response suppression on the growth of MDA-MB-231 cells, and the IC_50_ (50% inhibition concentration) of diamond NPs was determined to be 37 µM [86]. More recently, distinct ND-mediated delivery systems were investigated to enhance efficacy and decrease the side effects of anticancer drugs. It has been shown that monosaccharides can alter and construct targeted drug delivery systems [87]. Accordingly, Zhao et al. synthesized a glycopolymer-ND-amonafide (ND-Polymer-AMF) delivery system and evaluated its antiproliferative effect on MCF-7 and MDA-MB-231 cell lines. They found that at lower concentrations of amonafide, cell viability was inhibited. They also discovered that ND-Polymer-AMF could enhance the dispersibility of amonafide in the aqueous solution and also fructose moieties on the outer surface of ND-Polymer-AMF could bind to GLUT5, which is an overexpressed sugar transporter on the plasma membranes of cancer cells [88]. In another recent work, amphiphilic polymer-coated NDs were synthesized to encapsulate azonafide. The results revealed enhanced cytotoxicity of encapsulated azonafide in MCF-7 cells [89]. The antiproliferative role of PTX in the combination of NDs on MDA-MB-231cells was also proved, i.e., it has been shown that PTX+NDs can promote mortality of MDA-MB-231 cells, in addition to those mortality effects generated by NDs or PTX alone [86]. Likewise, other investigations demonstrated the effective role of ND-mediated delivery systems in various breast cancer models [90,91,92]. Some works showed negligible cytotoxicity of NDs against several cells with different origins [93,94], and work by other researchers indicated the cytotoxic activity of NDs in a concentration-dependent manner [86,95]. The pathway experiments reveal that the internalization of NDs in cancer and non-cancer cells occurs through the same pathway, i.e., energy-dependent and receptor-mediated endocytosis (RME) pathways; however, the internalization numbers of NDs are considerably more in cancer cells than in non-cancer cells [96]. Despite controversial findings in the literature, safety concerns, particularly regarding the impacts of NDs on healthy/normal cells, are still big challenges in the application of ND-based nanomaterials in breast cancer therapy. In other words, although multifunctional NDs could be a good candidate for the delivery of numerous therapeutic agents, research into NDs and their impact as a delivery system for therapeutic agents is still in its infancy. Thus, more investigation must be carried out to pave the way for their application in clinical trials and, ultimately, for improving patient outcomes.

## 6. Silisium (Silicon)-Based Nanomaterials

Silisium is well known as the earth crust’s second-most abundant element, behind oxygen, supplying low-cost and rich resource support for numerous silicon-based applications. By virtue of its outstanding mechanical and semiconductor properties, silisium materials dominate the electronics industry and act as the prominent semiconductor materials to date. [97]. To date, silicon materials of numerous nanostructures (nanodots, nanoparticles, nanorods, nanoribbons, and nanowires) have been manipulated. Silicon nanoparticles (SiNPs) and silicon nanowires (SiNWs) are identified as the most significant zero- and one-dimensional silicon nanostructures, respectively. Consistently, SiNPs are utilized in pharmaceutical technology. Conversely, SiNWs could serve as a platform for SERS studies [98]. In a recent study, the organotin(IV) complex and chlorambucil (two different cytotoxic agents) were loaded on fibrous silica NPs and then decorated with folic acid and Alexa Fluor 647. The study demonstrated enhanced cytotoxicity of two chemotherapeutic agents against MDA-MB-231 cells as well as a higher cell migration inhibition [99]. A successful encapsulation of breast cancer-specific gene 1-small interference RNA (BCSG1-siRNA) in chitosan-silicon dioxide NPs was reported by CUI et al. The authors observed the cytotoxic effect of chitosan-silicon dioxide/BCSG1-siRNA NPs on the growth of MCF-7 cells. They observed that the designed BCSG1-siRNA plasmid could selectivity and significantly downregulate BCSG1 gene expression. Thus, their nanomaterials demonstrated considerable antitumor influences in breast cancer cells [100].

Usually, three types of Si-based nanomaterials can be utilized in nanopharmaceutical research: mesoporous SiNPs (MSNs), periodic mesoporous SiNPs (PMONPs), and porous SiNPs (pSiNPs) (Figure 4). Recently, MSNs were proposed as imaging or drug delivery agents [101]. MSNs have a well-defined internal mesoporous structure (2–10 nm diameter) and a large pore volume (0.6−1 cm^3^/g). The shape, surface, and pore sizes of MSNs can be custom designed, offering several unique possibilities for the loading of antitumor agents. Many studies examined the cytotoxicity of MSNs loaded with DOX; for instance, the anticancer effects of DOX-MSNs@hyaluronic acid-gelatin-PEG on MDA-MB-231 cells by upregulation of the MMP-2 mechanism [102], niclosamide-DOX-COOH-Chi-MSNs on MDA-MB-231, SK-BR-3, and MCF-7 cells by inhibition of Wnt/*β*-catenin signaling mechanism [103], and MSN-siRNA/Aptamer@DOX on MDA-MB-231 cells by downregulation of TIE2 (tyrosine kinase with immunoglobulin-like and EGF-like domains 1) mechanism [104] have been reported. In addition, the antiproliferative effects of MSNs loaded with other anticancer agents have been studied. Konoplyannikov et al. prepared MSNs loaded with salinomycin and showed that this nanomaterial has a major cytotoxic effect on MCF-7/MDR1 cells (breast cancer and multidrug-resistant cells). Thus, their work revealed that salinomycin-loaded MSNs could be utilized for moderate chemotherapy of both primary cancer tumors and metastasis [105]. In a more recent study, Mohan Viswanathan et al. observed that prepared gallium nitrate and curcumin complex loaded on MSNs (GaC-HMSNAP) could significantly reduce the growth of MCF-7 cells by increasing PARP (Poly (ADP-ribose) polymerase), GSK 3β(S9), cleaved caspase-6, caspase-9, and caspase-6 expression. Furthermore, GaC-HMSNAP reduced mitochondrial proteins; for example, SOD1 (superoxide dismutase 1), HSP60 (heat shock protein 60), and prohibitin1. Consequently, their results showed that GaC-HMNSAP could provoke cell death through the mitochondrial intrinsic cell death pathway [106].

pSiNPs could also offer a platform for the vectorization of anticancer agents, especially hydrophobic agents, into the pores. Landgraf et al. loaded camptothecin (a highly cytotoxic chemotherapeutic compound) into pSiNP (160 nm) that was conjugated with EGFR-targeting antibody cetuximab. The authors revealed that the CPT-pSiNP-anti-EGFR decreased the growth of MDA-MB-231BO cells, increased survival rate, reduced primary tumor growth, and decreased metastases in a mouse model [107].

Consequently, most of the reported cases revealed that silica-based nanomaterials could be utilized in breast cancer therapy and combination therapy. Thus, developing simpler methods to synthesize silica-based nanomaterials is a high priority, as is enhancing their cancer- targeting abilities. Furthermore, the existing research about silica-based nanomaterials’ combination therapies is immature. Surely, chemotherapy-based nanosystems applying silica-based nanomaterials have a brilliant adaptable future and excellent potential for clinical translation.

## 7. Germanium-Based Nanomaterials

Germanium (Ge) is a metalloid with a semiconductor feature and is nonessential for human health. Ge exists in plants, animals, and soil as a natural compound. Ge possesses a diversity of interesting properties, for instance, narrow bandgap, high charge carrier mobility, high refractive index, and high ion intercalation capacity. Organic Ge and its compounds have the potential to enter the human body through the respiratory tract, digestive tract, skin, blood circulation system, and other ways. In October 2003, the FDA rejected adding organic germanium to human supplements because it had caused nephrotoxicity (kidney damage) and death in humans, even when people were healthy. The American Cancer Society also determined that organic Ge and its compounds have the potential to interfere with drugs and consequently are harmful to humans. It further warned that the usage of organic Ge alone could have serious health consequences for conventional cancer care [108,109,110].

Studies into the use of Ge nanomaterials in biological applications have been neglected. Ma et al. showed that water-soluble GeNPs (less than 10 nm) have high toxicity to cells. These GeNPs initiated cell necrosis by elevating intracellular calcium concentration, which increases ROS levels [111]. Hence, the intrinsic poor biodegradation of most Ge nanomaterials may prevent their further in vivo applications and clinical translation in biomedicine. However, there have been few reports available on Ge nanomaterials’ application in biology. Amongst these studies, one includes the study of Bezuidenhout et al., who revealed that there are no adverse cytotoxic effects of self-seeded Ge nanowires on L929 (murine aneuploid fibrosarcoma) and MCF-7 cells [112]. In 2016, McVey et al. compared the synthetic protocols of Si and Ge nanocrystals (NCs). They found that Si and Ge NCs possess low toxicity, so they could be used in biomedical applications [113]. In 2021, Ge et al. also synthesized a novel type of 2-D germanene nanosheets (GeH NSs) with excellent oxidative biodegradability, which has excellent degradability and biocompatibility [114]. Accordingly, Ge-based nanostructures have received considerably less attention, probably owing to their complex chemistry and the lack of convenient/predictable methods for their preparation. Therefore, we know very little about the behavior of Ge nanomaterials after treatment with cancer cells, specifically, breast cancer. Thus, we ought to have a deeper understanding of the exact mechanisms of these types of nanomaterials with cancer cells.

## 8. Tin-Based Nanomaterials

According to archaeological reports, people from ancient civilizations began utilizing bronze (an alloy that mostly contains tin and copper) for the design of durable and harder weapons and tools. Therefore, the earliest manipulation of tin (Sn) is during the bronze age. Today, tin is utilized for the manufacture of tin cans for the canning of processed food [115]. The oxide form of tin is the common form of tin. Sn(II)O (stannous oxide) and Sn(IV)O_2_ (stannic oxide) are the two primary oxides of tin. Tin oxide (SnO_2_) is a critical metal oxide semiconductor with a stable n-type wide bandgap (3.6 eV at 300 K). SnO_2_ belongs to the category of surface-sensitive materials, i.e., when molecules get adsorbed on the surface, charge transfer between the frontier orbitals of the adsorbates and the support surface may occur. Diverse morphologies of low-dimensional SnO_2_ nanostructure have been described as 0-D NPs, 1-D nanorods, nanobelts, nanowires, nanotubes, and 2-D nanosheets. Furthermore, recently, three-dimensional (3-D) hierarchical architecture has also been reported by a few researchers. Moreover, the doping of SnO_2_-based nanomaterials offers a convenient way to tailor their electrical, structural, and optical properties [116]. SnO_2_ has been employed extensively in various fields, including chemical sensors, environmental monitoring, solar cells, catalysis, and leakage detection [117]. However, despite their widespread use in many fields, in vitro molecular studies evaluating the safety/toxicity issues of SnO_2_ nanomaterials are limited. In 1929, the earliest experiments on the cytotoxicity of Sn compounds against mouse cancer were performed. Some 40 years later, various Sn(IV) compounds were surveyed for their in vivo antitumor activity against some cancer cells in mice. However, many of them were found to be inactive against other solid tumor cells [118]. Between 2016 and 2021, several articles were published on the effects of Sn NPs on breast cancer cells. For example, Guo et al. demonstrated that SnO_2_ NPs (30 nm) can prevent the proliferation of MCF-7 cells with mitigation of mitochondrial membrane potential, deactivation of catalase and superoxide dismutase activity, downregulation of p-PI3K/p-AKT/p-mTOR, and overexpression of the Bax/Bcl-2 signaling pathway [119]. Ahamed et al. observed the same results when they explored the effects of SnO_2_ NPs (21 nm) in MCF-7 cells. They reported that SnO_2_ NPs can induce toxicity in MCF-7 cells via oxidative stress [120]. The anticancer effects of SnO_2_ NPs (40 nm) extracted from *Rheum emodi* root on MDA-MB-231 cells were also examined and the potential cytotoxic effects of these NPs were proved [121]. In another study, undoped SnO_2_ and cobalt-doped NPs were biosynthesized and the significant in vitro anticancer and antioxidant effect of these NPs on MCF-7 cells was demonstrated [122]. Zhai et al. showed that the green synthesized SnNPs (18.13 nm) could induce cytotoxicity in MCF-10, MCF-7, and Hs 319.T (breast infiltrating ductal cell carcinoma) [123]. There is a great potential for Sn nanomaterials and future investigations about Sn on the nanoscale could provide some more encouraging results in drug delivery systems. It is anticipated that Sn NPs would give rise to the design of better antitumor agents with improved specificity.

## 9. Lead-Based Nanomaterials

Throughout cell evolution, about 24 (biogenic) metal species have been selected and assigned biological functions based on their intrinsic physicochemical properties and bioavailability. The most frequently found biogenic metal ions are Na^+^, K ^+^, Mg ^2+^, Ca^2+^, Zn^2+^, Mn^2+^, Fe^2+/3+^, Co^2+/3+^, Ni ^2+^, and Cu^+/2+^, which play vital roles in a multitude of essential tasks such as protein structure stabilization, enzyme catalysis, blood coagulation, signal transduction, muscle contraction, hormone secretion, taste and pain sensation, respiration, and photosynthesis. Other abiogenic/xenobiotic metal species, e.g., Hg^2+^, Pb^2+^, Al^3+^, or Tl^+^/Tl^3+^, upon entering organisms lacking efficient defensive mechanisms against such intruders, can adversely impact the cellular processes by competing with the native cations in various proteins.

Lead (Pb^2+^), with an unknown biological function in higher organisms, is commonly found in soil and water at concentrations ∼1000-fold higher than its natural levels, a consequence of utilizing leaded fuel, lead-containing water pipes, and leaded paint. This metal has harmful impacts on human health, as manifested through severe neurological, cardiovascular, reproductive, renal, endocrine, hematological, and/or immune dysfunctions. This is mainly because lead is a potent neurotoxin that interferes with signaling cascades in the brain, causing cognitive and psychiatric disorders [124]. Due to its strong toxicity, several government agencies have set strict regulations on the maximum permitted Pb levels. For instance, the maximum Pb concentration in drinking water defined by the World Health Organization (WHO) is 0.01 mg L^−1^ (48 nM) [125,126,127]. Some reports have indicated that high levels of heavy metals (lead, selenium, mercury, and cadmium) can accumulate in cancerous breasts and their presence can be one of the causes of cancer initiation [128]. Pb can activate estrogen receptor α (ERα) to direct the estrogen target genes’ expression and breast cancer cell reproduction [128]. There is only limited information about lead oxide nanoparticles (PbONPs) regarding their toxicological impacts [129]. Thus, although experiments on the mechanism concerning pb-based nanomaterials in breast cancer are limited, it seems that pb nanomaterials can induce oxidative stress in cells [129]. Collectively, although a few articles showed the biocompatibility of their synthesized Pb NPs [130], there are no articles to date about the effects of nanolead on the growth of breast cancer cells.

## 10. Mechanisms of Cell Death

Before nanomaterials (herein the elements of G14 nanomaterials) reach the outer membranes of tumor cells, they should interact with the microenvironment around the tumor cells [131]. The microenvironment, such as extracellular matrix and fibrosis, can alter the properties of nanomaterials and modify their interactions with the cell membrane and their intracellular fate [131]. Furthermore, the zone of nanomaterial–cell interactions can be influenced by the interplay of several microenvironmental factors, including MMPs, prostaglandins, vascular endothelial growth factor (VEGF), and bradykinin [132]. The pH of the tumor microenvironment can also alter the nanomaterial–cell interactions and the entry of nanomaterials [133]. Once nanomaterials reach the exterior membrane of a tumor cell, they can interact with the extracellular matrix or components of the plasma membrane and enter the cell. It has been demonstrated that nanomaterials can enter the tumor cell mainly through endocytosis pathways [134]. The five main mechanisms of endocytosis by which nanomaterials can enter cells are phagocytosis, caveolin-mediated endocytosis, clathrin-mediated endocytosis, macropinocytosis, and clathrin/caveolae-independent endocytosis (Figure 5). Other entry mechanisms are direct microinjection, disruption of the cell membrane, hole formation, and passive diffusion [132,134]. It should be noted that the physicochemical properties of nanomaterials, such as shape, size, surface charge, hydrophobicity, and surface functionality, can affect their cellular uptake [132].

Intracellular trafficking of nanomaterials has a crucial role in the cellular fate of nanomaterials and their therapeutic efficacy. After the entry of nanomaterials into tumor cells via endocytic vesicles, generally, their fate is determined by the intracellular trafficking/sorting mechanisms, mostly mediated by a network of cellular endosomes in conjunction with the lysosomes, endoplasmic reticulum (ER), and Golgi apparatus [135]. Several hidden factors affect the cellular trafficking of nanomaterials, including protein corona and cell vision. The set of proteins that bind to the surface of the nanomaterial is referred to as the protein corona, i.e., what cells “see” is corona-coated nanomaterials rather than their pristine surfaces [135]. Cell vision indicates the behaviors/mechanisms that each cell can employ in response to nanomaterials [136]. These two issues can alter the intracellular trafficking of nanomaterials.

### 10.1. Most Common Mechanisms of Cell Death Induced by Nanomaterials

When nanomaterials enter tumor cells could interfere with cell components, for instance, mitochondria, and generate damage affecting their functions. In response to stress, mitochondria can produce ROS [137]. Thus, following exposure to nanomaterials, the intracellular generation of ROS can rapidly and sharply increase. ROS are chemically reactive particles that contain oxygen, including hydroxyl radicals (^•^OH), reactive superoxide anion radicals (O^2−^), and hydrogen peroxide (H_2_O_2_) [138]. The excess ROS induced by nanomaterials can affect the organelles and biomolecules (DNA/RNA, lipids, and protein) structures of tumor cells, which causes cell death [137,139,140]. Furthermore, overburdened ROS can initiate further irreversible cell damage such as the inactivation of cell receptors, the leakage of the organelles’ contents, cell cycle arrest, and the rupturing of the membranes of organelles [141,142]. It has been shown that during mitosis, nanomaterials could interact with chromosomes and cause breaks into chromosomes or disturb the process of mitosis, by chemical binding or mechanically. Nanomaterials could also bind or interact with DNA molecules during interphase and cause a variety of DNA strand breakages and DNA damage. They also can influence DNA replication and transcription of DNA into RNA [137].

Nanomaterials can also target signaling pathways and control cell capacities. All distinctive features of cancer cells are mediated through signaling pathways that have become deregulated. Currently, 12 signaling pathways that perform a vital function in cancer growth have been recognized, including RAS, cell cycle apoptosis, Notch, Hedgehog signaling (HH), transcriptional regulation, APC, signal transducers and activators of transcription (STAT), DNA damage control, transforming growth factor-beta (TGF-β), a mitogen-activated protein kinase (MAPK), phosphatidylinositol-3 kinase (PI3K), and chromatin modification [143,144]. Among these pathways, the relationship between kinase pathways and cancer has attracted the attention of several scientists. Thus, several nanomaterial-based inhibitors were designed for various families of kinases, for instance, the mTOR, receptor tyrosine kinases (RTKs), MAPK, and PI3K. The RTKs’ signaling cascades affect the motility of cells, mitogenesis, cell survival and differentiation, as well as gene expression [144,145]. One of the highly important RTK subfamilies in cancer is the ErbB or EGFR family. Three major intracellular signaling cascades stimulated by EGFR activation include the MAPK pathway, the PI3K pathway, and the antiapoptotic Akt/protein kinase B (PKB) pathway [144]. The activity of PI3K is crucial for cellular responses to malignant transformation and growth factors. The PI3K/mTOR network has detected actionable target proteins in breast cancers. mTOR refers to two protein complexes, mTORC1, and mTORC2, that have a fundamental role in integrating signals from nutrients and growth factors to monitor metabolism, cell cycle progression, and protein synthesis [146]. In the following, we will discuss the most common mechanisms of cell death induced by G14 nanomaterials. It should be noted that after loading of antitumor agents, the mechanism of cell death may differ.

### 10.2. Mechanism of Cell Death Induced by Carbon-Based Nanomaterials

It has been reported that carbon-based nanomaterials can enter the cell, usually by adhesive interactions, and are found free in the cytoplasm; i.e., this group of nanomaterials has the potential to interact directly with the cytoskeleton to influence mechanotransduction. As a typical example, graphene-based nanomaterials have a great affinity with the cellular membrane [24]. It has been confirmed that the van der Waals interaction and hydrophobic interaction are the main driving force in GO–membrane contact [147]. The extraction of phospholipids through this interaction could induce serious membrane damage and even cell death. It should be noted that any modification on the surface of graphene-family nanomaterials could change their cellular uptake; for example, positively charged graphene sheets could enter MCF7 cells by receptor-mediated endocytosis and phagocytosis [148]. When graphene-family nanomaterials pass through the cell membranes of breast cancer, cell death is initiated by various mechanisms. One of the toxicological processes proposed for numerous graphene-based nanomaterials is ROS generation [149,150]. Graphene-based nanomaterials can also enter the nucleus and directly interact with DNA, mainly through π–π stacking and hydrogen bonding, causing DNA distortion and even DNA cleavage [151]. The large GQDs have a tendency to stick to the ends of the DNA molecule, causing the DNA to unfold; however, the small GQDs easily enter the DNA molecule, leading to DNA base mismatch [152]. Based on recent research, rGO could induce apoptosis in breast cancer cells by mitochondrial membrane potential reduction, deregulation of mitochondrial proteins, activation of caspase-9 and caspase-3, cell cycle arrest, and deregulated P21 expression (Figure 6). rGO also caused autophagy in breast cancer cells [153]. Graphene-based nanomaterials could also affect the cell signaling pathways. GQDs could downregulate the expressions of P-glycoprotein, multidrug resistance protein 1 (MRP1) [154]. It has been reported that the synthesized nanocomposite consisting of rGO incorporated with Cu (rGO-Cu) could downregulate MMP-9, cathepsin D, and Bcl-2 gene expression and upregulate P53 gene expression [25]. In breast cancer, MMP9 and cathepsin D are involved in cell invasion and metastasis through the hydrolysis of fibronectin, proteoglycans, and collagens by their lysosomal aspartic protease activity [155]. The Bcl-2 protein family can regulate the permeability of the mitochondrial inner membrane and also can mitigate the induction of apoptosis, whereas Bax causes permeability of the outside mitochondrial membrane, which releases soluble proteins into the cytosol, where they stimulate the activation of caspase [156].

The literature review showed that few surveys had been conducted to assess the mechanisms of cell death induced by CNTs in breast cancer. Recent studies revealed CNTs utilize either an endocytic pathway or passive diffusion to penetrate through cellular membranes [157]. Endocytosis is also the proposed mechanism for translocating CQDs and NDs into breast cancer cells. Nevertheless, some research demonstrated this process can be carried out through non-endocytic pathways; thus, the internalization mechanism theory for these nanomaterials still needs more investigation [158]. Based on recent studies NDs (0-D) and CNTs (1-D) show higher uptake rates than sheet-like GO (2-D). Thus, there is a positive relationship between shape, size, and surface modifications of carbon-based nanomaterials and cellular uptake. Prior studies have illustrated that when CNTs pass through cells, they can initiate cell death by downregulation of Bcl-2 and upregulation of Bax [59], mitochondria damage (induced by SWCNTs), and membrane integrity (induced by MWCNTs) (Figure 3) [53]. It has been demonstrated that the mechanism of CNTs in cells could be related to the purity, size, fabrication route, length, structure, surface chemistry, and concentration of CNTs. Based on recent research, CQDs could induce apoptosis in breast cancer cells by cell cycle arrest in the G0/G1 phase, DNA damage [82], ROS generation, LDH release [80], and activation of caspase-3 [80,81] (Figure 7). Collectively, the mechanism of cell death induced by carbon-based nanomaterials in cancerous cells is different. As an example, compared with GO and NDs, CNTs induce a dramatic release of LDH owing to their needle-like structure, which could be more mobile and can more easily penetrate the cell membrane, causing greater cell membrane damage. Furthermore, CNTs and GO induce a high level of ROS. In contrast, NDs show the ability to scavenge ROS [159]. It should also be mentioned that carbon-based nanomaterials tend to interact with a variety of proteins in biological fluids, forming a protein corona on the surface of nanomaterials. Thus, the interaction of carbon-based nanomaterials with proteins can affect the cellular uptake and mechanism of cell death.

### 10.3. Mechanism of Cell Death Induced by Silisium-Based Nanomaterials

Recently, by utilizing the pharmacological inhibitors of major endocytic pathways, the uptake mechanisms of SiNPs in MCF-7 cells were demonstrated. The results proved that the uptake of SiNPs, especially amino-functionalized SiNPs, is strongly affected by actin depolymerization. Thus, F-actin plays a crucial role in the process of internalization of SiNPs. More importantly, positively charged SiNPs have higher cell uptake than negatively charged SiNPs, which is crucial in designing drug carriers that can cross the mucosal barriers and enable a noninvasive delivery of biological therapeutics [160]. It has also been shown that when nano-SiO_2_ passes through the cell membranes of breast cancer, it can downregulate EGFR, proto-oncogene c-Src, and signal transducer and activator of STAT3 phosphorylation. Furthermore, nano-SiO_2_ could reduce the expression of survivin, cyclins, fibronectin, and focal adhesion kinase (FAK) (Figure 8A). Thus, it could be proposed that nano-SiO_2_ could induce apoptosis in breast cancer cells by disruption of EGFR dimerization, modulation of downstream signaling cascades, and disruption of cellular adhesion, migration, and invasion [161]. Results of other investigations also indicate that mesoporous silica-based nanomaterials (<100 nm diameter) could not disorganize the actin filaments in the cytoskeleton; nevertheless, rod-shaped MSNs could disorganize and disrupt the actin filaments with poorly formed filament bundles in the area near the cell membrane and at the edges of lamellipodia and filopodia [162]. Such cytoskeleton destruction causes higher cellular penetration of rod-shaped MSNs, leading to more serious damage to the cytoskeleton. It should be noted that since MSNs demonstrate tunable pore size, porous interior, and large surface area, they could act as an excellent reservoir for various drug molecules and other pharmaceutical materials of interest. Various targeting moieties have been also utilized on the surface of MSNs, for example, targeting folate receptor, transferrin receptor, VEGF receptor, mannose receptor, HER2 receptor, and c-type lectin receptor [3] (Figure 8B). However, in terms of the biological point of view, the clinical application of MSNs is limited, owing to the rapid clearance of nanoparticles by immune and excretory systems after administration. Another limitation of using MSNs in clinical applications is that upon administration of MSNs in the body and exposure to blood, protein corona could form on the surface of the MSNs, which can eventually block the pores and reduce the release of drugs from the pores of MSNs. Therefore, the detailed in vivo analysis of pharmacokinetic and pharmacodynamic experiments, possible immunogenicity, and rigorous biodistribution of MSN-based systems ought to be utilized before aiming to translate clinically [163].

### 10.4. Mechanism of Cell Death Induced by Tin-Based Nanomaterials

The results of published articles indicate that when Sn-based nanomaterials (SnO_2_ NPs or SnNPs) pass through the cell membranes, cell death is initiated by several mechanisms. One of the toxicological processes proposed is ROS generation [119,120,122]. Sn-based nanomaterials could also induce apoptosis in breast cancer cells by mitigation of mitochondrial membrane potential, deactivation of catalase and superoxide dismutase activity, downregulation of PI3K/AKT/mTOR, overexpression of Bax/Bcl-2 signaling pathway [119], LDH leakage, as well as by the accumulation of cells in the sub-G1 phase [120]. Consequently, it could be speculated that Sn-based nanomaterials hinder the expression of cytokines or signaling transduction pathways based on activating PI3K, which could mitigate the downstream AKT kinase function and relevant reactions stimulated by AKT. Inactivation of the PI3K/AKT/mTOR pathway leads to overexpression of proteins, including Bax, and down-expression of Bcl-2, and finally leads to apoptosis (Figure 9) [119,144].

## 11. Conclusions and Future Prospects

Nowadays, cancer nanotechnology brings several perspectives, beyond traditional breast cancer treatment, that are more efficacious and potentially safer. Numerous nano-based systems have been approved by the FDA or are in clinical trials for cancer treatments. From the prior discussion of the latest literature, it is obvious that there are still several barriers to overcome before the widespread clinical use of innovative nanodrugs for human breast cancer. One is to enhance the encapsulation efficiency and loading concentration of therapeutic agents. A therapeutic concentration of a drug is necessary to attain anticancer effects. The other important item is to select a biocompatible and nontoxic nanomaterial. Nanomaterial that goes through blood circulation ought to eventually clear from the body.

A concern that stems from systemic circulation is the safety of targeting ligands, which must be nontoxic and biocompatible. Furthermore, the physicochemical characteristics of nanomaterials (colloidal stability, surface modification, size, structure, and composition) all impact the clinical outcome. Any small batch-to-batch difference in nanomaterials’ characterization can initiate dramatic variations in their biocompatibility and toxicity in vitro and in vivo. In this review, we attempted to highlight the application of nanomaterials of the elements of the G14 in breast cancer treatment to clarify whether these substrates are useful or not in nanopharmacy. In writing this review, we found that unequivocal statements were hard to find, as a result of the different influences of this group of nanomaterials on breast cancer cell lines. The intensity of the induced cell death varies depending on the type of nanomaterials; i.e., Ge and, somehow, Pb nanomaterials exhibit a high level of toxicity even in healthy cells, whereas diamond and Si/SiO_2_ nanomaterials exhibit a low level of toxicity. According to the literature, luminal subtypes have a better prognosis and are less aggressive than basal-like and claudin-low tumors. Based on our survey, the elements of G14 nanomaterials (especially the graphene family) can affect both less and more aggressive subtypes (Figure 10).

It should be noted that the mechanisms of tumor cell death induced by the elements of G14 nanomaterials are dissimilar. For example, GO and modified GO particles have a great affinity with the cellular membrane, and adhesion to the cell surface could induce their cytotoxicity [24]. Other research proved that the elements of G14 nanomaterials induce cell death by increasing oxidative stress and ROS formation [63,75,120], downregulating the critical genes’ expression [69,100] and blocking or overexpressing various signaling pathways [68,96,103]. Various studies proved that the elements of G14 nanomaterials are favorable for use in drug delivery systems [45,50,65,76]. It should be mentioned that in the area of anticancer agents’ development, the literature is limited, making it difficult to reach conclusions about the potential effects of nanomaterial of the element of G14 on breast cancer cells. Even so, a close inspection of these conclusions makes it clear that some chemical and physical parameters could affect the antiproliferative effects of nanomaterials on breast cancer cell lines. It is also vital to note that in many researches, the oxides and other salt forms of nanomaterials of the elements of the G14 have been examined. Therefore, materials’ characteristics have an essential role in antiproliferative effects and cannot be ignored.

It should be recalled that cancer is dynamic and depends on tumor type and stages. Thus, permanent treatments do not always work, and resistance to anticancer agents arises. Therefore, we need more multifunctional nanodrug designs that can overcome this problem. We ought to be certain of the ideal nanomaterial for breast cancer treatment while considering patient variations, and in every nanomaterial design, the mechanism of action ought to be recognized. Although the long-term effects of these kinds of nanomaterials on humans need to be defined, we believe this review will fundamentally provide new perspectives for the application of G14-based nanomaterials in breast cancer therapy. It also reveals the importance of taking these groups of nanomaterials from the lab to the clinic to test their efficacy and safety in humans.

## Figures and Tables

**Figure 1 pharmaceutics-14-02640-f001:**
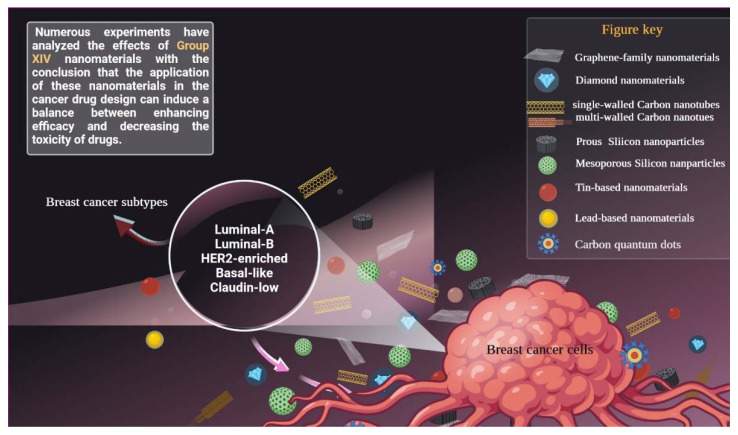
Nanomaterials derived from G14 whose antiproliferative effects on breast cancer subtypes were established.

**Figure 2 pharmaceutics-14-02640-f002:**
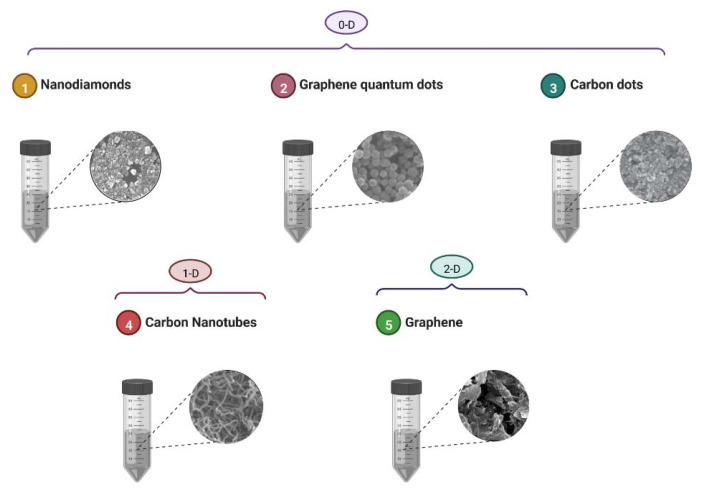
Nanoforms of carbon-based materials with examples for 0-D, 1-D, and 2-D carbon nanostructures.

**Figure 3 pharmaceutics-14-02640-f003:**
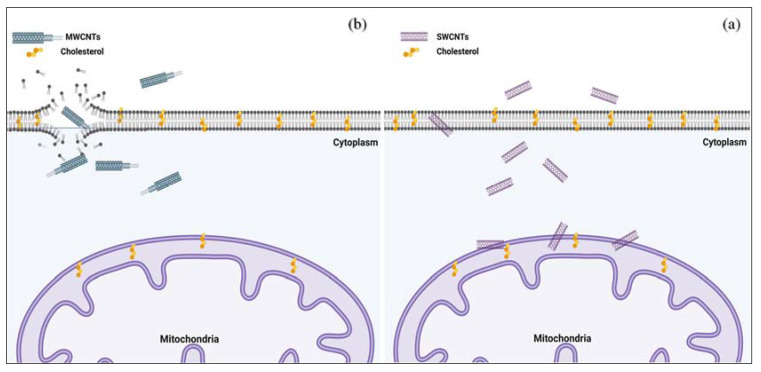
Schematic description of interaction of SWCNTs (**a**) and MWCNTs (**b**)with cell membrane and mitochondria (based on reference [53]).

**Figure 4 pharmaceutics-14-02640-f004:**
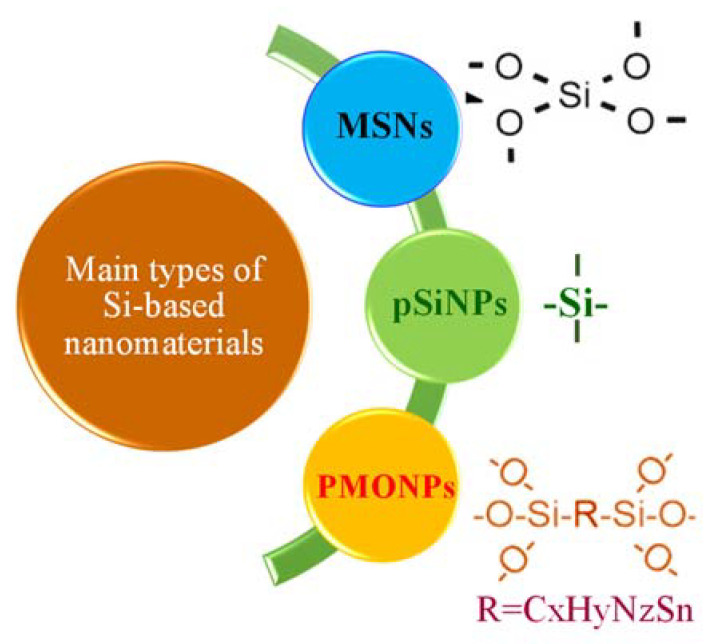
The chemical structure of the three main types of Si-based nanomaterials: PMO, MSN, and pSiNP.

**Figure 5 pharmaceutics-14-02640-f005:**
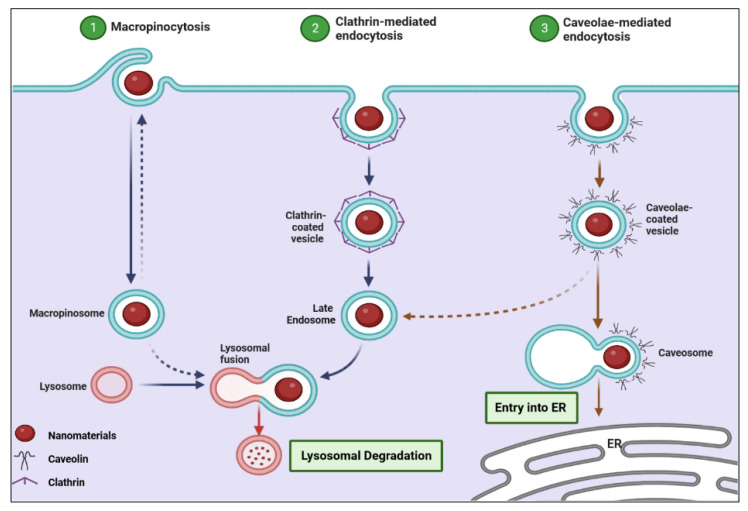
Three main mechanisms of endocytosis by which nanomaterial can enter cells. Abbreviation: ER—endoplasmic reticulum.

**Figure 6 pharmaceutics-14-02640-f006:**
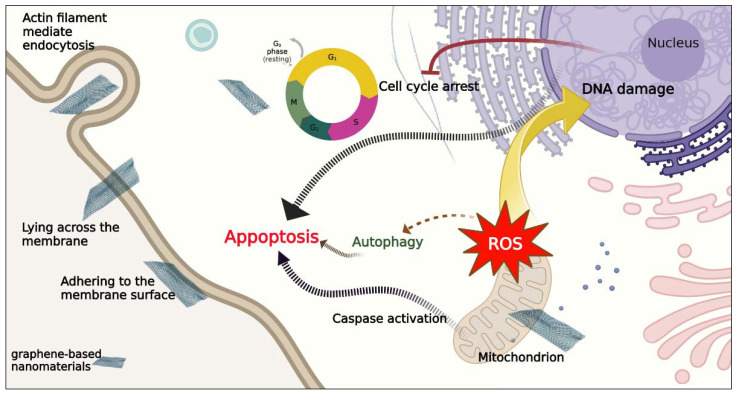
Schematic model to illustrate the proposed mechanism of cell death induced by graphene-based nanomaterials on oxidative stress, apoptosis, cell cycle, and autophagy in breast cancer cells, based on recent studies [151,152,153].

**Figure 7 pharmaceutics-14-02640-f007:**
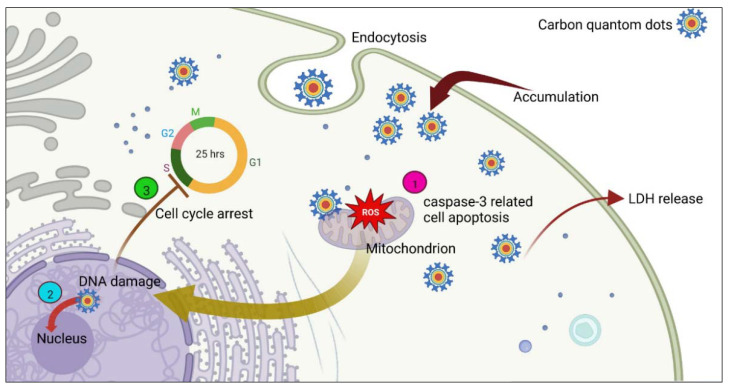
Schematic model to illustrate the proposed mechanisms of cell death induced by CQDs in breast cancer cells, based on recent studies [79,80,81,82]. (1) Increase in ROS levels, (2) DNA damage, (3) cell cycle arrest.

**Figure 8 pharmaceutics-14-02640-f008:**
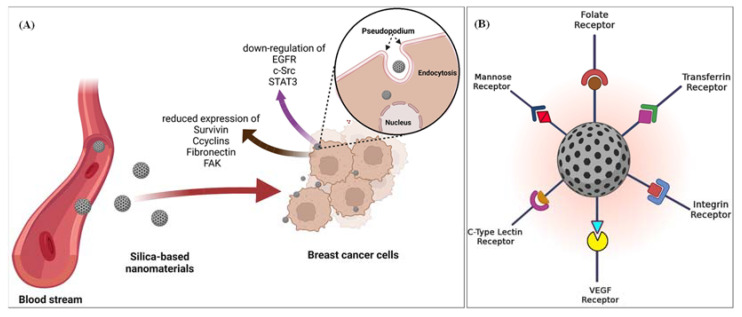
(**A**) Schematic model to illustrate endocytic mechanisms and the proposed mechanisms of SiNPs in breast cancer, based on recent studies [160,161]. (**B**) Probable surface modifications of MSNs to the overexpressed receptors in the cancer microenvironment.

**Figure 9 pharmaceutics-14-02640-f009:**
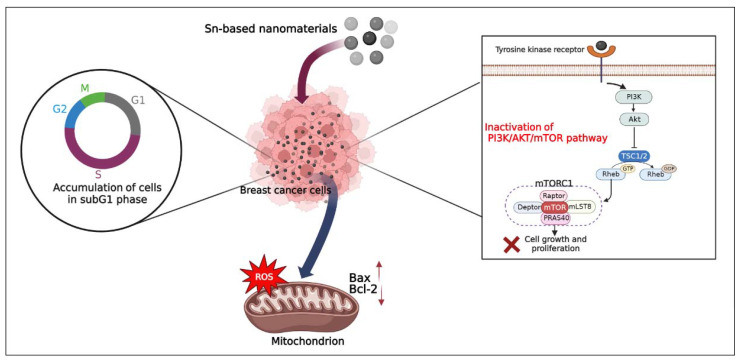
Schematic model to illustrate the proposed mechanisms of cell death induced by Sn-based nanomaterials in breast cancer, based on recent studies [119,120,122].

**Figure 10 pharmaceutics-14-02640-f010:**
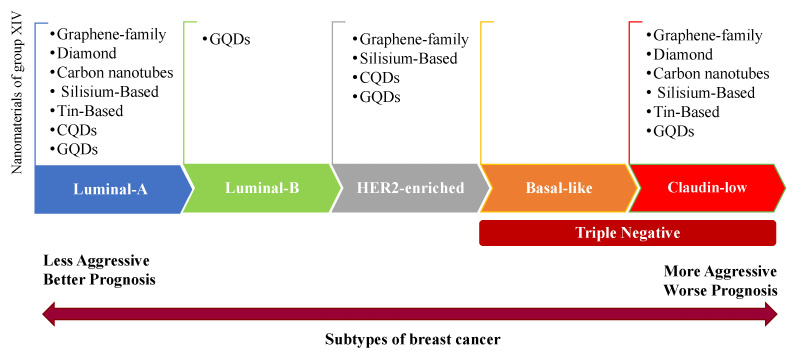
A brief classification of the elements of the G14 nanomaterials applied in different subtypes of breast cancer, based on recent studies. Based on the literature, the elements of G14 nanomaterials can affect both less and more aggressive subtypes.

**Table 1 pharmaceutics-14-02640-t001:** Breast cancer subtypes, molecular information, and permanent cell lines established from them *.

Molecular Subtype		Luminal A	Luminal B	HER2-Enriched	Basal-Like	Claudin-Low
cell lines		MCF-7 MDA-MB-134 MDA-MB-175 MDA-MB-415 T-47D	BT-474 MDA-MB-330 MDA-MB-361	AU-565 HCC-1008 HH-315 HH-375 KPL-4 MDA-MB-453 SK-BR-3 SK-BR-5 SUM190PT UACC-893	BT-20 CAL-148 HCC1143 KPL-3C MDA-MB-468 SUM229PE	BT-549 MDA-MB-157 MDA-MB-231 SK-BR-7 SUM102PT SUM1315M02 SUM149PT SUM159PT
molecular signatures	**ER ^1^**	P ^§^	P	N	N	N
	**PR ^2^**	N/P ^#^	N/P	N	N	N
	**HER2 ^3^**	N ^‡^	P	P	N	N

* Data from ATCC (American Type Culture Collection) and Expasy (Cellosaurus); ^1^ ER signifies estrogen receptor; ^2^ PR signifies progesterone receptor; ^3^ HER2 signifies human epidermal growth factor receptor2; ^§^ signifies positive; ^‡^ signifies negative; ^#^ signifies positive in some cell lines and negative in some cell lines.

**Table 2 pharmaceutics-14-02640-t002:** Studies evaluating the therapeutic impact of CNT complexes on breast cancer cells.

Material	Anticancer Agent/Drugs	Cellular/Animal Models	Outcomes	Ref.
**SWCNTs**	PEG-*β*-estradiol-Lobaplatin	MCF-7 C57BL/6 mice	- Inhibition of MCF-7 cell growth - Reduction of Lobaplatin side effects - No histopathological variations in organs	[67]
**CNTs**	Ginsenoside Rg3	MDA-MB-231 BT-549	- Apoptosis induction in cells - Suppressed the PD-1/PD-L1 pathway	[68]
**MWCNTs**	Cisplatin	MDA-MB-231	- Reduction of cells growth - Significant reduction of P53 and caspase-3 expression - Downregulation of NF-κB in cells	[69]
	Mesoporous silica-Chitosan-DOX	MCF-7 Rats	- DOX released at pH 4.0 - Higher plasma concentration of DOX - A lower volume of distribution and clearance	[70]
	DOX/Metformin	MCF-7 4T1	- Marginal beneficial impact on 4T1 cells	[71]
	PEGylated Carboplatin	MDA-MB-231	- No significant impact on the viability of cells - Exhibit pH-responsive drug activity in a sustained way particularly at pH 6.8	[72]

**Table 3 pharmaceutics-14-02640-t003:** Studies evaluating the therapeutic impact of CQDs derived from biomolecules and medicinal plants on breast cancer cells.

**Precursors**	**Size (nm)**	**Cellular Models**	**Outcomes**	**Ref.**
*Aloe barbadensis miller*	3.2	MCF-7	- Apoptosis induction in cells	[79]
Ginsenoside Re	4.6	MCF-7	- Increase in LDH release - Increase in ROS levels - Caspase-3-related cell apoptosis	[80]
Walnut Oil	12	MCF-7	- Caspase-3-related cell apoptosis	[81]
*Nerium Oleander*	2.05	MCF-7	- Influence on the cell cycle - DNA damage	[82]

**Precursors**

## Data Availability

Not applicable.

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
