# Peer review of "Recent Advances in Nanomaterials of Group XIV Elements of Periodic Table in Breast Cancer Treatment"

_pharmaceutics, 2022, doi:10.3390/pharmaceutics14122640_

Round 1

Reviewer 1 Report

Hekmat and co-workers reported the review of “Recent Advances in Nanomaterials of Group-XIV Elements of Periodic Table in Breast Cancer Treatment”, which the results have been presented correctly, and the contents fall well into the scope of the journal.

Author Response

We appreciate your review and your comment.

Reviewer 2 Report

The review should be improved with other nanomaterials based on elements from group XIV of the periodic table, such as carbon quantum dots and graphene quantum dots, as an example, but not only:

1. Soroush Moasses Ghafary, Elnaz Rahimjazi, Hadiseh Hamzehil, Sayed Mostafa Modarres Mousavi, Maryam Nikkhah, Saman Hosseinkhani, Design and preparation of a theranostic peptideticle for targeted cancer therapy: Peptide-based codelivery of doxorubicin/curcumin and graphene quantum dots, Nanomedicine: Nanotechnology, Biology, and Medicine 42 (2022) 102544

2. Junlong Liang, Qianwei Huang, Chenxiang Hua, Jinhua Hu, Biling Chen, Junmin Wan, Zhiwen Hu, Bing Wang, pH-Responsive Nanoparticles Loaded with Graphene Quantum Dots and Doxorubicin for  Intracellular Imaging, Drug Delivery and Efficient Cancer Therapy,(2019), DOI: 10.1002/slct.201803807 

3. Rahul S. Tade and Pravin O. Patil, Theranostic Prospects of Graphene Quantum Dots in Breast Cancer, (2020), https://dx.doi.org/10.1021/acsbiomaterials.0c01045

4. Sheetal Devi, Manish Kumar, Abhishek Tiwari, Varsha Tiwari, Deepak Kaushik, Ravinder Verma, Shailendra Bhatt, Biswa Mohan Sahoo, Tanima Bhattacharya, Sultan Alshehri, Mohammed M.  Ghoneim, Ahmad O. Babalghith and Gaber El-Saber Batiha, Quantum Dots: An Emerging Approach for Cancer Therapy, (2022) DOI: 10.3389/fmats.2021.798440

Author Response

The review should be improved with other nanomaterials based on elements from group XIV of the periodic table, such as carbon quantum dots and graphene quantum dots, as an example, but not only:

  1. Soroush Moasses Ghafary, Elnaz Rahimjazi, Hadiseh Hamzehil, Sayed Mostafa Modarres Mousavi, Maryam Nikkhah, Saman Hosseinkhani, Design and preparation of a theranostic peptideticle for targeted cancer therapy: Peptide-based codelivery of doxorubicin/curcumin and graphene quantum dots, Nanomedicine: Nanotechnology, Biology, and Medicine 42 (2022) 102544
  2. Junlong Liang, Qianwei Huang, Chenxiang Hua, Jinhua Hu, Biling Chen, Junmin Wan, Zhiwen Hu, Bing Wang, pH-Responsive Nanoparticles Loaded with Graphene Quantum Dots and Doxorubicin for Intracellular Imaging, Drug Delivery and Efficient Cancer Therapy,(2019), DOI: 10.1002/slct.201803807 
  3. Rahul S. Tade and Pravin O. Patil, Theranostic Prospects of Graphene Quantum Dots in Breast Cancer, (2020), https://dx.doi.org/10.1021/acsbiomaterials.0c01045
  4. Sheetal Devi, Manish Kumar, Abhishek Tiwari, Varsha Tiwari, Deepak Kaushik, Ravinder Verma, Shailendra Bhatt, Biswa Mohan Sahoo, Tanima Bhattacharya, Sultan Alshehri, Mohammed M. Ghoneim, Ahmad O. Babalghith, and Gaber El-Saber Batiha, Quantum Dots: An Emerging Approach for Cancer Therapy, (2022) DOI: 10.3389/fmats.2021.798440

Thank you for your valuable comments.

Two sections about carbon quantum dots and graphene quantum dots were added. The suggested papers also were added.

Pages 6&7 lines 248-386: “Newly, among all graphene derivatives, graphene quantum dots (GQDs) are considered a newly emerging material, which represents a class of 0-D material with quasi-cyclic crystal structures. QDs could be a good source of light spanning from ultraviolet to infrared [38]. GQDs have a strong quantum property and are acquired from cutting a graphene monolayer into small pieces in dimensions of 2-20 nm. There are various reports about different applications of GQDs in medicine and pharmacy for example bioimaging, drug delivery, biosensors, and optical detectors. GQDs revealed good biocompatibility as well as the lowest cytotoxicity, although surface functionalization helps in their biodegradation [39,40]. The remarkable biological properties of GQDs, highlight the superiority of GQDs over modified graphene or GO [41]. In the recent era, numerous structural characteristics of GQDs have been analyzed by several researchers. GQDs and graphene oxide quantum dots (GOQDs) show excellent photoluminescence properties which are predominantly attributable to surface states. Accordingly, Ahirwar et al. synthesized GQDs/GOQDs (with a size range of 1.5–5.5 nm) as a photosensitizer and used 365 nm ultraviolet tube light as an irradiation source. They observed that GQDs/GOQDs were uptake by cells easily and induced very low cytotoxicity towards MCF-7 cells. However, when 365 nm UV was performed on cells treated with GQDs/GOQDs more than 90% of cells were killed during a short exposure of 5 min [42]. It should be noted that although, photodynamic therapy using GQDs is very easy and effective for cancer treatment. Nevertheless, this therapy will be limited to skin-related cancers as the depth of penetration of ultraviolet irradiation is extremely less. GQDs can also utilize in drug delivery systems. In a recent study, GQDs and MiRGD peptides were utilized for the targeted delivery of curcumin and DOX. The in vivo study on breast cancer-bearing BALB/c mice showed that the prepared DOX-curcumin-MiRGD-GQDs peptideticles could be considered suitable multifunctional theranostic peptideticles for targeted drug delivery and tracking [43]. Liang et al. prepared pH-responsive nanoparticles loaded with GQDs and DOX (GQDs@DOX/PB). Their experiments demonstrated that GQDs@DOX/PB could promote the release of DOX in a mild acidic microenvironment [44]. Other investigations exhibited the effective role of anticancer drug-loaded on GQDs NPs in various breast cancer models. For example, the anticancer effects of DOX-loaded zeolitic imidazolate framework-GQDs NPs on 4T1 cells under near-infrared irradiation at 1.5 W/cm2 [45], DOX-CuS@GQDs NPs on MDA-MB-231 cells under high temperature and NIR laser irradiation at 1 W/cm2 [46], Herceptin-β-CD-GQDs-HER on MCF-7 and BT-474 cells [47], Methotrexate-GQDs on MCF-7 cells [48], DOX-GQDs-mesoporous silica-PEG on MDA-MB-231 [49], and DOX-β-cyclodextrin-SS-GQDs on MDA-MB-231 and SK-BR-3 cells [50] were approved. Collectively, numerous experiments have indicated that the most invasive types of breast cancer cells can be killed efficiently with GQDs [48,49,51]. However, these experiments should be further conducted in clinical trials and animal models for realizing their potential and recognizing their safety and efficacy profile”.

Pages 9&10 lines 351-400: “Quantum dots (QDs) are made from very small metal particles and in their crystal core, there are around 100–100,000 atoms In ultraviolet light, these dots glow, and the shade of QDs is limited by their size. [38]. Owing to their tiny size, QDs perform differently than bulk solids due to quantum-control phenomena [72]. Recently, QDs have gotten lots of attention. Some of the most emerging QDs are carbon QDs (CQDs), MXenes QDs (a class of 2-D inorganic compounds that consist of carbides, nitrides, and carbonitrides), MBenes QDs (consist of 2-D transition metal borides), and metal oxide QDs (such as ZnO QDs and MgO QDs) [38]. CQDs which discovered by researchers in 2004 [73], are novel 0-D carbon-based nanomaterials known for their relatively effective fluorescence characteristics, large-scale preparation, good water solubility, small size, chemical stability, and ease of surface functionalization [73]. Various investigations have highlighted that CQDs are useful for numerous cancer treatment with or without combination with irradiation. CQDs also can effectively be utilized in drug delivery systems by combining them with antitumor drugs. For example, Li et al., designed the 5-aminolevulinic acid-CQDs -Glu-β-cyclodextrin system (5-ALA-CQD-Glu-β-CD) and loaded with DOX. The in vitro study revealed that the produced system had morphological abnormalities and substantial cytotoxicity against MCF-7 cells. They also proposed that the generation of ROS upon 635 nm (for 15 min) laser irradiation improved the therapeutic effects of their designed system [74]. Samimi et al., synthetized the nitrogen doped CQDs-quinic acid-gemcitabine (N-CQD-quinic acid-Gem). Then the growth of MCF-7 cells after treatment with the synthetized system was examined. Moreover, the tissue biodistribution profiles and blood circulation of the resulting N-CQDs-quinic acid-Gem were examined following intravenous administration through the tail vein of mice. The synthetized system demonstrated high tumor accumulation and great luminescent properties. They proposed that the nanosized system could penetrate in the tissue system, assist drug uptake by cells, permit an effective drug delivery, and ensure action at the targeted location [75]. The cytotoxicity of DOX-transferrin-CQDs on MCF-7 cells was investigated by Mahani et al. Their investigation revealed that their designed nano system diminished cell viability efficiently. The nanocarrier also revealed a pH-dependent DOX release behavior. The inhibitory effect of the nanocarrier could be due to its overcome on multidrug resistance of cancer cells and also targeting by conjugated transferrin on the nanosized system surface which considerably improved the delivery of DOX into the cancer cells through the overexpressed transferrin receptors [76]. Collectively, based on previous publication CQDs could improve drug absorption in the body and drug bioavailability, enhance drug contact criteria, and increase the medication’s impact. CQDs also could also overcome obstacles such as solubility of drug, lack of drug selectivity, as well as drug resistance. CQDs can interact with the cancer cells resulting in the formation of ROS, which can kill cancer cells. However, it should be noted that the traditional chemical methods make use of harsh and toxic chemical additives for the synthesis of CQDs that offer risks to the biological environment [38,77]. Thus, various researchers tried to synthetize CQDs from biological materials. For example, Malavika et al., constructed amorphous CQDs utilizing Aloe barbadensis extract as a precursor. Their investigation showed that CQDs have apoptosis effects on MCF-7 cancer cells. It has been also informed that the CQDs synthesized from ginsenoside Re [78] and walnut oil [79] were found to have anticancer property against MCF-7 cells. Overall, even though CQDs have made substantial contributions to tumor therapy, numerous challenges restrict their clinical applications in human owing to the poor biocompatibility. consequently, more biocompatible, biodegradable, and one-step synthesis methods ought to be discovered. Furthermore, a probable genotoxic effect of CQDs must be studied extensively”.

Reviewer 3 Report

1. The mechanism should be elaborated.

2. 5-8 figures and tables should be provided for easy understanding.

3. References are not fully cited, such as: 

European Journal of Medicinal Chemistry, 2019, 182,111612.

Journal of Controlled Release, 2018, 278, pp. 122-126.

Author Response

  1. The mechanism should be explained.

For each nanomaterials the mechanisms that the authors suggested in their publication were highlighted.

Page 6 lines 204, 210-213, 223-224, 245-246

Page 7 lines 272-274

Page 8 lines 309,317-320, 322-324, 331-339

Page 9 Table 2

Page 10 lines 368, 376-378, 385-389

Page 11 lines 420-422, 431-435

Page 12 lines 463-466, 474-480, 486-493

Page 13 lines 524-526

Page 14 lines 565-574

Page 15 lines 606-607

Also, all mechanism were collected in the Conclusion and Future Prospects part: “It should be noted that the mechanisms of tumor cell death induced by the elements of G14 nanomaterials are dissimilar. For example, GO and modified GO particles have a great affinity with the cellular membrane, and with adhesion to the cell surface could in-duce their cytotoxicity [24]. Some other research proved that the elements of G14 nanomaterials induce cell death by increasing oxidative stress and ROS formation [63,74,116], down-regulating the critical genes expression [68,96], and blocking or overexpressing various signaling pathways [67,92,99]. Furthermore, various studies proved that the elements of G14 nanomaterials are favorable for using in drug delivery systme [45,50,65,75].” (Page 16 lines 641-648)

  1. figures and tables should be provided for easy understanding.

Thank you for your valuable comments.

All tables were rewritten, and all figures were drawn again. Furthermore, 3 other figures were added:

Page 5, Figure 2: “Several nanoforms of carbon-based materials with examples for 0-D, 1-D, 2-D, and 3-D carbon nanostructures”.

Page 9, Figure 3: “Schematic description of interaction of MWCNTs and SWCNTs with cell membrane and mitochondria. Cell membrane reveals higher cholesterol distribution than mitochondria membrane and consequently offers more hydrophobic microenvironment. MWCNTs with lower surface tension results in membrane integrity, however SWCNTs with greater surface tension leads to mitochondria damage.

Page 11, Figure 4: “The antiproliferative effects of 0.4 and 0.5 µM PTX with 10, 15 and 20 µM NDs. Cell death increased to 52% when 0.5 µM PTX was utilized in the presence of 20 µM NDs. Based on these observations it seemed that PTX combined with NDs could promote mortality of cells besides those mortality effects induced via PTX or NDs alone”.

Furthermore, the format of Tables 1 and 2 changed for easy understanding.

  1. References are not fully cited, such as: 

European Journal of Medicinal Chemistry, 2019, 182,111612.

Journal of Controlled Release, 2018, 278, pp. 122-126.

Thanks for your suggestion, these references were added (Page 8 line 325 (Reference [66]); Page 11 line 415 (Reference [85])).

Reviewer 4 Report

The work, entitled “Recent Advances in Nanomaterials of Group-XIV Elements of Periodic Table in Breast Cancer Treatment” aims to review the recent investigations about the antiproliferative effects of nanomaterials of the elements of the group XIV in the periodic table on breast cancer cells. It is a compressive and important review concerning the recent findings. A perspective was also included. I really enjoy read it. The paper is well written and has the makings of a publication.

Author Response

(The authors gave the same response as above.)

Round 2

Reviewer 2 Report

This form of the review is acceptable from my point of view.

Author Response

(The authors gave the same response as above.)

Reviewer 3 Report

The research on mechanism is not deep enough.

Author Response

In section 10, the mechanisms of cell death  induced by group 14 nanomaterials in breast cancer were discussed (Pages 14-20).

Also for each group of nanomaterials, the proposed mechanisms of cell death induced in breast cancer based on recent studies were drawn.
